# Specific Isolation of *Clostridium botulinum* Group I Cells by Phage Lysin Cell Wall Binding Domain with the Aid of S-Layer Disruption

**DOI:** 10.3390/ijms23158391

**Published:** 2022-07-29

**Authors:** Zhen Zhang, François P. Douillard, Hannu Korkeala, Miia Lindström

**Affiliations:** Department of Food Hygiene and Environmental Health, Faculty of Veterinary Medicine, University of Helsinki, 00014 Helsinki, Finland; zhen.zhang@helsinki.fi (Z.Z.); francois.douillard@helsinki.fi (F.P.D.); hannu.korkeala@helsinki.fi (H.K.)

**Keywords:** *Clostridium botulinum*, phage lysin, cell wall binding domain, S-layer, diagnostics, flow cytometry, magnetic separation

## Abstract

*Clostridium botulinum* is a notorious pathogen that raises health and food safety concerns by producing the potent botulinum neurotoxin and causing botulism, a potentially fatal neuroparalytic disease in humans and animals. Efficient methods for the identification and isolation of *C. botulinum* are warranted for laboratory diagnostics of botulism and for food safety risk assessment. The cell wall binding domains (CBD) of phage lysins are recognized by their high specificity and affinity to distinct types of bacteria, which makes them promising for the development of diagnostic tools. We previously identified CBO1751, which is the first antibotulinal phage lysin showing a lytic activity against *C. botulinum* Group I. In this work, we assessed the host specificity of the CBD of CBO1751 and tested its feasibility as a probe for the specific isolation of *C. botulinum* Group I strains. We show that the CBO1751 CBD specifically binds to *C. botulinum* Group I *sensu lato* (including *C. sporogenes*) strains. We also demonstrate that some *C. botulinum* Group I strains possess an S-layer, the disruption of which by an acid glycine treatment is required for efficient binding of the CBO1751 CBD to the cells of these strains. We further developed CBO1751 CBD-based methods using flow cytometry and magnetic separation to specifically isolate viable cells of *C. botulinum* Group I. These methods present potential for applications in diagnostics and risk assessment in order to control the botulism hazard.

## 1. Introduction

*Clostridium botulinum* is a Gram-positive, obligately anaerobic, spore-forming bacterium that produces botulinum neurotoxin (BoNT), the most potent biological toxin known. *C. botulinum* spores are resistant and widely distributed in nature. Once the spores are disseminated in food or feed, or in the gastrointestinal tract or in the wounds of humans and animals, in favorable conditions, the spores may germinate into vegetative BoNT-producing cultures and cause botulism, a potentially life-threatening neuroparalytic disease [1]. Because of the high heterogeneity of *C. botulinum* strains, secondary taxonomic groups are conventionally used in their classification [2]. Human botulism is predominantly associated with *C. botulinum* Groups I and II, whereas animal botulism is mainly associated with *C. botulinum* Group III. Each group also includes phylogenetically related but nontoxinogenic strains or species. *C. botulinum* Group I possesses two distinct clades, one consisting mainly of toxinogenic *C. botulinum* Group I strains and the other consisting mainly of nontoxinogenic *C. sporogenes* strains. Both clades also harbor sporadic nontoxinogenic and toxinogenic strains, respectively [3,4]. To address the two clades together, we hereafter refer to them as *C. botulinum* Group I *sensu lato*.

Laboratory diagnostics of botulism are primarily based on the detection of BoNT in clinical samples [5]. Further confirmation of the diagnosis should be achieved through the identification and isolation of viable *C. botulinum* cells, but this is often hampered by the lack of suitable microbiological tools [6]. Moreover, safety monitoring regarding *C. botulinum*-contaminated in foods and feeds warrants novel strategies for the isolation of *C. botulinum* [7]. The conventional isolation procedures are based on culture in a non-selective enrichment broth or on solid media and colony screening for BoNT production and biochemical characteristics, such as the lipase and lecithinase activity [8]. However, the biochemical markers are nonspecific, and standardized BoNT assays utilize animals. The isolation of *C. botulinum* is thus time-consuming, laborious, and often fails. Efficient methods for the specific identification and isolation of *C. botulinum* are warranted.

As a result of long-term co-evolution with bacteria, bacteriophages exhibit a specific affinity for bacteria. Phage lysins, the key components required for host bacterial lysis, show rigid host specificity by their cell wall binding domains (CBD) recognizing unique cell wall epitopes present in specific bacterial species or serovars [9]. This feature highlights the potential of phage lysins as antibacterial agents or as diagnostic tools for specific bacteria [10,11]. A number of phage lysins have been characterized for various bacterial species, including *C. botulinum* Group I [12] and Group II [13]. Furthermore, the host range and recognition mechanisms of several lysin CBDs have been characterized [14,15,16,17]. Lysin-CBD-based methods have been established for the detection and isolation of specific pathogenic bacteria, such as *Listeria monocytogenes* [18,19,20], *Bacillus anthracis* [21], *Bacillus cereus* [22,23], *Staphylococcus aureus* [24,25], and *Clostridium perfringens* [26]. Recently, we characterized the first antibotulinal phage lysin, CBO1751, which shows a considerably higher lytic activity against *C. botulinum* Group I *sensu lato* than the other tested bacterial strains [12]. The rigid host specificity of CBO1751 led us to hypothesize that its CBD, when isolated from the lytic domain, could be used as a biological probe for the specific isolation of viable cells of *C. botulinum* Group I. A recent report shows a number of CBO1751 homologs (annotated as N-acetylmuramoyl-L-alanine amidase) in both *C. botulinum* Group I and *C. sporogenes* to be targets of endogenous CRISPR systems [4], which could indicate that CBO1751 and its homologs constitute promising candidates for specifically targeting *C. botulinum* Group I and *C. sporogenes*.

Here, we expressed the CBD of CBO1751 and characterized its binding affinity. The CBD of CBO1751 was bound specifically to *C. botulinum* Group I *sensu lato* strains. In some Gram-positive bacteria, the peptidoglycan cell wall is surrounded externally by a para-crystalline proteinaceous layer, termed the surface layer (S-layer). The S-layer fulfils a wide range of biological functions, including acting as a protective sheath of cells [27]. We found that some, but not all, *C. botulinum* Group I strains possess an S-layer and the presence of the S-layer significantly impaired the recognition of CBO1751 CBD to cells. We employed acid glycine treatment to disrupt the S-layer, which facilitated the binding of CBO1751 CBD to all of the tested *C. botulinum* Group I strains. Combining the S-layer disruption with CBO1751-CBD-based flow cytometry separation and magnetic separation, we could specifically isolate viable *C. botulinum* Group I cells.

## 2. Results

### 2.1. Enhanced Binding of CBO1751 CBD to Clostridium botulinum ATCC3502 Cells by S-Layer Disruption

To examine the binding of CBO1751 CBD to *C. botulinum* cells, two fluorescent proteins, mCherry and mTagBFP, were used to tag the predicted CBO1751 CBD, generating fusion proteins mCherry-CBD and mTagBFP-CBD, respectively (Figure 1A). Through recombinant expression in *Escherichia*
*coli*, mCherry-CBD and mTagBFP-CBD proteins were obtained and subjected to binding analysis with vegetative cells of *C. botulinum* Group I ATCC3502 and ATCC19397. Intriguingly, only sporadic cells of ATCC3502 were labeled on the cell surface with red fluorescence (mCherry-CBD) and blue fluorescence (mTagBFP-CBD). In contrast, practically all *C. botulinum* ATCC19397 cells were labeled efficiently with mCherry-CBD and mTagBFP-CBD (Figure 1B). These observations suggest that CBO1751 CBD recognizes cell wall epitopes of *C. botulinum* ATCC19397 cells, but fails with most ATCC3502 cells. CBO1751 is known to have a robust lytic activity against both strains [12]. Given that *C. botulinum* ATCC3502 is phylogenetically closely related to ATCC19397 (96.4% genomic similarity), it is unlikely that *C. botulinum* ATCC3502 has different cell wall epitopes from those of ATCC19397. Therefore, we speculate that some components localized on the cell surface of ATCC3502 cells might interact with cell wall epitopes and hinder the binding of CBO1751 CBD.

The bacterial S-layer is the peripheral proteinaceous component of the cell surface. S-layers have been detected in *C. botulinum* Group I strain 190L and Group II strain Saroma [28,29]. The S-layer of ATCC3502, if it exists, might interfere with the binding of phage lysin CBD to the cell wall epitopes. To examine this possibility, we treated vegetative cells of ATCC3502 with an acid glycine buffer, which is a routine procedure to remove S-layer proteins from bacterial cells [30], and then performed the binding analysis with mCherry-CBD and mTagBFP-CBD. Contrary to the observation with non-treated ATCC3502 cells, the cells treated with 0.2 M glycine-HCl (pH 3 or 4) were all strongly labeled with mCherry-CBD and mTagBFP-CBD (Figure 1B and Appendix A). We further explored the S-layer of *C. botulinum* using transmission electron microscopy (TEM). In Figure 2A, the S-layer is shown as the outermost peripheral structure of the cell surface of *C. botulinum* ATCC3502. After treatment with glycine-HCl (pH 3), the S-layer was disrupted and detached from the cell wall. After treatment with glycine-HCl (pH 4), the S-layer became granular and appeared to be aggregated to some extent. The combined observations indicate that disruption of the S-layer by acid glycine treatment strongly enhances the binding of CBO1751 CBD to *C. botulinum* ATCC3502 cells. Moreover, our microscopic examinations suggest that *C. botulinum* ATCC19397 does not possess an S-layer. Briefly, the ATCC19397 cells with or without acid glycine treatment were similarly labeled with both fluorescent CBO1751 CBD reporters (Figure 1B) and showed no S-layer in the TEM imaging (Figure 2B).

### 2.2. Binding Specificity of CBO1751 CBD

We further investigated the binding specificity of CBO1751 CBD in a range of *C. botulinum* and related strains as well as other species using mCherry-CBD. In the binding assays with seven phylogenetically diverse *C. botulinum* Group I strains of different neurotoxin serotypes, mCherry-CBD only showed binding to four of the tested *C. botulinum* Group I strains (Table 1 and Appendix A). After S-layer disruption with 0.2 M glycine-HCl (pH 4) for 1 min, all seven Group I strains were labeled markedly with mCherry-CBD. mCherry-CBD also showed binding to *C. sporogenes* NINF45, a member of the *C. sporogenes* clade [31]. In contrast, mCherry-CBD did not show binding to any of the four *C. botulinum* Group II strains, *C. botulinum* Group III BKT2873, *Clostridium baratii* CCUG24033, *Clostridium butyricum* BL86/13, *C. perfringens* ATCC13124, *B. cereus* ATCC14597, *Bacillus*
*subtilis* 1012M15, *L. monocytogenes* EGD-e, *S. aureus* ATCC12600, or *E. coli* 5 alpha, regardless of the acid glycine buffer treatment (Table 1 and Appendix A). The tested *C. botulinum* Group II and Group III strains exhibited autofluorescence emission when viewed with a Texas Red filter (Appendix A). The cell autofluorescence can be distinguished from a specific signal by its markedly lower intensity than that of mCherry-CBD-labeled cells (Appendix A). These observations suggest that the binding specificity of CBO1751 CBD is restricted to *C. botulinum* Group I *sensu lato*.

To test if the CBO1751 CBD can detect *C. botulinum* spores, we examined the binding of mCherry-CBD and mTagBFP-CBD to *C. botulinum* ATCC19397 in a sporulating late-stationary-phase culture. mCherry-CBD and mTagBFP-CBD showed binding to the sporulating cells with a weak florescence, but at the same time intensely labeled massive cell debris, leading to the spores being virtually indistinguishable from the cell debris (Appendix A). We concluded that CBO1751 CBD is not applicable for the isolation or identification of spores.

### 2.3. Flow Cytometry Separation of Clostridium botulinum Group I

To test if fluorescently labeled CBO1751 CBD can be used to specifically separate viable *C. botulinum* Group I cells from other bacterial cells, we performed fluorescence-based flow cytometry separation from a cell pool consisting of equal amounts of *C. botulinum* type A strain ATCC19397 (or ATCC3502), type E strain CB11/1-1, type F strain 202F, and type CD strain BKT2873 cells. MitoTracker Green (MTG) was applied to label all of the cells indiscriminately, whereas mTagBFP-CBD only labeled *C. botulinum* Group I cells. As shown in the flow cytometry analysis in Figure 3A, a subpopulation with a high intensity of mTagBFP-CBD (mTagBFP+, 25.8% of all cells) was identified in the cell pool containing ATCC19397 without acid glycine treatment. In the cell pool containing ATCC3502, an mTagBFP+ subpopulation accounting for 5.54% of all cells was detected without acid glycine treatment, whereas the population of mTagBFP+ subpopulation was 29.44% in the cell pool treated with 0.2 M glycine-HCl (pH 4) for 1 min. These findings suggest that S-layer disruption by acid glycine significantly enhanced mTagBFP-CBD-based cell separation. Moreover, we confirmed that treatment with glycine-HCl (pH 4) for 1 min does not significantly affect the viability of *C. botulinum* (Figure 3B).

After sorting, the cell populations with a high fluorescence intensity of mTagBFP-CBD were serially diluted for the most probable number (MPN) assay. Each strain was enumerated by PCR detection of genes encoding BoNT types A, E, F, and C/D in the dilution series. The PCR products specific for *bont/A* (101 bp), *bont/E* (389 bp), *bont/F* (543 bp), and *bont/CD* (327 bp) were detected in the cell pools before sorting (Figure 3C and Appendix A). After sorting, the *bont/A* fragment was consistently detected in all MPN dilution series of sorted ATCC19397 and ATCC3502 cells. On the contrary, the *bont/E* fragment was present only in the first ten-fold dilution, but not in further dilutions, and the *bont/F* and *bont/CD* fragments were not detectable in any of the MPN dilutions. The results suggest that *C. botulinum* Group I cells formed a vast majority (>99.9%) of the sorted cell populations, demonstrating that CBO1751 CBD-based cell sorting was efficient for the specific separation of viable *C. botulinum* Group I cells. 

Samples sorted from the cell pool containing ATCC3502 without acid glycine treatment showed growth only in the first ten-fold MPN dilution and tested PCR positive only for *bont/A*, suggesting that sporadic ATCC3502 cells were labeled and sorted by mTagBFP-CBD without acid glycine treatment. Microscopic examination showed that these sorted samples were comprised mainly of cell debris with a few intact cells with mTagBFP fluorescence (data not shown). These results are consistent with the above observations of sporadic non-treated ATCC3502 cells labeled with mTagBFP-CBD (Figure 1B). It is plausible to speculate that *C. botulinum* ATCC3502 cells might sporadically lose their S-layer structure under laboratory culture conditions. S-layer loss in the laboratory has been reported for some bacterial species [32]. Taken together, these observations suggest that S-layer disruption is a necessary step for the efficient flow cytometry separation of *C. botulinum* Group I cells.

### 2.4. Magnetic Separation of Clostridium botulinum Group I

We further explored the magnetic separation of viable *C. botulinum* Group I cells using superparamagnetic micro-sized beads. M-280 tosylactivated Dynabeads^®^ (Thermo Fisher Scientific, Vantaa, Finland) were used for coating with mCherry-CBD. After coating, red fluorescence was observed on the surface of the M-280 beads (Figure 4A). Microscopic examination also showed that mCherry-CBD-coated M-280 beads were bound to ATCC19397 vegetative cells after incubation with excess cells for 15 min and magnetic separation (Figure 4A).

To evaluate the capture efficiency of magnetic separation, 10^4^–10^5^ vegetative cells were incubated with an excess of mCherry-CBD-coated M-280 beads (~3.3 × 10^7^). After magnetic separation, the captured cells were enumerated by MPN. As mCherry-CBD can directly bind to *C. botulinum* ATCC19397 cells, ATCC19397 cells without acid glycine treatment were directly subjected to magnetic separation, which yielded an average capture efficiency of 11.0%, with values ranging from 10.0 to 12.9% (Figure 4B). A comparable capture efficiency for *C. botulinum* ATCC3502 cells (9.0%) was achieved only after treatment with 0.2 M glycine-HCl (pH 4) for 1 min, suggesting that S-layer disruption of ATCC3502 cells is necessary for the binding of mCherry-CBD-coated M-280 beads. In contrast, the capture efficiency for *C. botulinum* CB11/1-1, 202F, and BKT2873 cells remained consistently lower than 1%, regardless of the acid glycine treatment. The results suggest that CBO1751 CBD-based magnetic separation is a promising method for the rapid isolation of *C. botulinum* Group I.

## 3. Discussion

Phage lysin CBDs are responsible for recognizing cell wall epitopes that are unique to particular types of bacteria, and for directing the lysin enzymatically active domain (EAD) to hydrolyze the cell wall peptidoglycan, which leads to bacterial cell lysis. Several studies demonstrate that lysin CBDs can specifically bind to target bacterial species, suggesting great potential for use as diagnostic tools in bacterial detection [33,34]. The first antibotulinal phage lysin, CBO1751, consists of an EAD of N-acetylmuramoyl-L-alanine amidase on the N-terminal side and a CBD of bacterial Src homology 3 (SH3b) on the C-terminal side [12]. While the SH3b domain is distributed widely among many bacteria [35], CBO1751 CBD shares a very low sequence identity with those predicted in the genomes of *C. botulinum* Group II and III, and other species, suggesting that the SH3b domain of CBO1751 might have a rigid binding specificity to *C. botulinum* Group I. Here, we performed binding assays using fluorescent protein-tagged CBO1751 CBD and demonstrated the binding specificity of CBO1751 CBD to *C. botulinum* Group I *sensu lato*. We further proved the feasibility to use CBO1751 CBD in the specific separation of viable *C. botulinum* Group I cells.

The fluorescent protein-tagged CBO1751 CBD bound to *C. botulinum* ATCC3502 cells was effective only with an S-layer disrupting treatment. This suggests that the S-layer blocks lysin CBD from binding to the cell wall epitopes. To the best of our knowledge, this is the first piece of evidence that an S-layer directly blocks the binding of phage lysin CBDs. S-layers form the outermost components of the cell envelope in some bacteria and archaea [36]. In Gram-positive bacteria, S-layer proteins are directly anchored to the peptidoglycan cell wall, providing a protective barrier that limits the access of hostile environmental factors to the cell wall substrate [37]. Therefore, it is plausible that the blocking effect of S-layers against phage lysin CBDs is a prevalent mechanism.

Disruption of the S-layer should be considered as an effective step to enhance the sensitivity and specificity of lysin CBD-based detection of bacteria. Generally, S-layers can be removed from bacterial cells by a range of disrupting agents, including low-pH glycine-HCl buffer, high-ionic-strength lithium chloride (5M), detergents, denaturants, and proteases [30]. We initially tested several disrupting agents, but most of them substantially reduced the viability of *C. botulinum* ATCC3502 cells (data not shown). Treatments with lithium chloride (5M) or glycine-HCl with pH 3 facilitated the labeling of mCherry-CBD to ATCC3502 cells, but concomitantly led to a strong reduction in cell viability (Appendix A). As the S-layer is not an essential structure and bacterial viable cells may lose S-layers under laboratory culture conditions [32], it appeared feasible to develop a compromised protocol that can sufficiently disrupt the S-layer without jeopardizing cell viability. Indeed, when the glycine-HCl treatment was adjusted to pH 4 for only 1 min, the S-layer was only mildly disrupted (Figure 2B), without a significant reduction in cell viability (Figure 3B), and mCherry-CBD could bind to the ATCC3502 cells (Figure 1B). 

While the S-layer strongly interferes with the binding of CBO1751 CBD to the cell wall epitopes, the lytic activity of lysin CBO1751 is likely unaffected by the S-layer. Our previous study showed a 10 min lag time in the onset of cell lysis after CBO1751 was added in a *C. botulinum* ATCC3502 cell suspension [12], whereas the *C. botulinum* ATCC19397 cells were lysed readily (this work, Appendix A). These observations suggest that S-layer hinders but does not prevent phage lysins from lysing the target cell. One possible explanation is that an intact lysin possesses mobility on the cell surface [38] to optimize access to the cell wall epitopes underneath the S-layer.

Bacterial S-layers have been detected in several pathogenic clostridial species, including *Clostridioides difficile*, *Clostridium tetani,* and *C. botulinum* [36,39]. The S-layer proteins (SLPs) of *C. difficile* and *C. tetani* use tandem cell wall binding 2 (CWB2) domains to anchor the S-layer to the underlying cell wall [40,41]. Although the genes encoding *C. botulinum* SLPs still need to be verified, two putative CWB2 family proteins (CBO0378 and CBO0380) in the genome of *C. botulinum* ATCC3502 show a similar amino acid composition to the previously reported SLP of *C. botulinum* type A strain 190L [29], suggesting that CBO0378 and CBO0380 may be the SLPs of *C. botulinum* ATCC3502 [42]. BLAST analysis did not find any CBO0378 or CBO0380 homologs in *C. botulinum* ATCC19397, which is consistent with the microscopic observation of a lacking S-layer in ATCC19397. Our analysis indicates that some but not all *C. botulinum* Group I strains possess an S-layer. Despite its protective role against harsh environmental conditions, the S-layer is proposed to have a compartmentalizing function in regulating the release of macromolecules such as bacterial toxins [36]. In addition, roles in bacterial adhesion, biofilm formation, and interaction with the host immune system have been proposed [37]. Further studies on the formation and variation of the S-layer in *C. botulinum* strains may provide important information about its roles in the survival and pathogenicity of *C. botulinum*. 

With the aid of S-layer disruption, we confirmed the binding specificity of CB1751 CBD to *C. botulinum* Group I and its closely related species *C. sporogenes*, which is in line with our previous finding on the host specificity of the phage lysin CBO1751 [12]. The present results provide a proof of concept for the specific separation of *C. botulinum* Group I cells from other bacteria with mTagBFP-CBD-based flow cytometry. This approach offers a powerful tool for the specific isolation of viable *C. botulinum* Group I cells. Recently, lysin CBD-based flow cytometry was used to detect *Staphylococcus* in the blood with a high sensitivity and specificity [43]. Further studies on flow cytometry application using mTagBFP-CBD might favor the development of high-throughput tools for complex matrices such as food and clinical specimens. Magnetic separation has become a routine laboratory method for the detection and isolation of pathogenic bacteria [44]. We demonstrated the application of mCherry-CBD-coated superparamagnetic M-280 beads in the magnetic separation of *C. botulinum* Group I cells with an average capture efficiency 9–11%. A wide range of capture efficiencies (10–90%) have been reported for magnetic separation assays depending on the applied detection antibodies, peptides, or phage lysin CBDs [45]. It is difficult to compare efficiencies between different magnetic separation approaches as no standardized protocols are available. It is worth noting that a very low non-specific capture efficiency (<1%) was achieved in our test, supporting a high specificity for the isolation of *C. botulinum* Group I cells. 

An ideal detection or isolation method for spore-forming bacteria would simultaneously consider both vegetative cells and spores. It is unclear if bacterial spores possess receptors for phage lysin CBDs on their surface in the same manner that vegetative cells do. As spores are assumingly not killed by bacteriophages and their outer layers are very different from those of vegetative cells, we primarily assumed that the CBO1751 CBD-based methods are not feasible for *C. botulinum* spores. While we did observe strong binding of fluorescent protein-tagged CBO1751 CBD on the surface of *C. botulinum* Group I spores in an aging ATCC19397 culture, cell debris was also labeled (Appendix A). On the contrary, only a very weak signal was observed on the purified spores (data not shown). These findings suggest that the fluorescent signal detected around mature spores was an artefact due to the signal derived from lysed vegetative cells. This supports our assumption that the detection or isolation of spores with the aid of phage lysin CBDs is not feasible. However, even unspecific binding of fluorescently tagged CBO1751 CBD to the spore surface, for example in enrichment cultures, might provide a serendipitous advantage in attempts to isolate *C. botulinum* from complex matrices.

In conclusion, we have expressed and characterized the active CBD of CBO1751 that can specifically bind to *C. botulinum* Group I *sensu lato* cells. The binding of CBO1751 CBD to cell wall epitopes is strongly affected by the presence of an S-layer. With the combination of S-layer disruption and fluorescent protein-tagged CBO1751 CBD labeling, we demonstrated efficient isolation of viable *C. botulinum* Group I cells using cell sorting and magnetic separation. Considering the specificity of CBO1751 CBD to *C. botulinum* Group I *sensu lato*, detection of the neurotoxin or its encoding gene should complement these approaches to exclude *C. sporogenes* or other nontoxinogenic Group I isolates. In summary, these approaches present the potential for the development of novel tools for diagnostics and risk assessment of botulism. 

## 4. Materials and Methods

### 4.1. Bacterial Strains and Culture

The bacterial strains are listed in Appendix A. All Clostridia were cultured in anaerobic Tryptone Peptone Glucose Yeast Extract (TPGY) medium at 37 °C or 30 °C (*C. botulinum* Group II strains) in an anaerobic workstation with an atmosphere of 85% N_2_, 10% CO_2_, and 5% H_2_ (MK III; Don Whitley Scientific Ltd., West Yorkshire, UK). *B. subtilis* 1012M15, *B. cereus* ATCC14579, *L. monocytogenes* EGD-e, *S. aureus* ATCC12600, and *E. coli* cells (Merck Millipore, Darmstadt, Germany) were grown in aerobic Luria–Bertani (LB) medium at 37 °C. When appropriate, growth media were supplemented with 100 μg/mL ampicillin and 34 μg/mL chloramphenicol.

### 4.2. Expression and Purification of mCherry-CBD and mTagBFP-CBD

Two DNA fragments each encoding a fluorescent protein fused to CBO1751 CBD were designed by inserting DNA encoding the amino acid residues 169–253 of CBO1751 (GenBank accession number CAL83288) to the 3′ end of the DNA encoding mCherry (AAV52164) or mTagBFP (AZQ25074), termed mCherry-CBD and mTagBFP-CBD, respectively. The DNA fragments were commercially synthesized and cloned in the NheI/SalI restriction sites of pET21b to generate a C-terminal fusion construct with a 6 × His tag (Biomatik Corporation, Ontario, Canada). The vectors were transformed into *E. coli* Rosetta 2 (DE3) pLysS cells (Merck Millipore). The expression of the His-tagged protein was induced with 1 mM IPTG at 30 °C for 8 h. The expressed protein was purified using metal-chelate affinity chromatography with Ni–IDA resin (Merck Millipore), as previously described [46]. The eluted proteins were dialyzed against Tris buffer (500 mM NaCl, 50% glycerol, 20 mM Tris–HCl, pH 7.9) for long-term storage or a PBS buffer (pH 7.4) for immediate use in coupling to magnetic beads.

### 4.3. Fluorescence Microscopy

Bacterial strains were cultured to the exponential growth phase with an optical density at 600 nm (OD_600_) of 0.5. In addition, late-stationary-phase cultures (3 and 7 days) were used to detect spores. Cells from 1 mL of culture were harvested by centrifugation at 10,000× *g* for 1 min and then washed gently and resuspended in 100 µL of PBS buffer. When S-layer disruption was required before fluorescent protein labeling, the cells were treated with 0.2 M glycine-HCl (pH 3 or 4) for 1 min and then centrifuged immediately at 10,000× *g* for 1 min. The separated cells were washed gently and resuspended in 100 µL of anaerobic PBS buffer. Prior to microscopic examination, the cells were mixed with mCherry-CBD or mTagBFP-CBD (final concentration, 1 µM) and incubated at room temperature for 3 min, followed by gentle washing with anaerobic PBS buffer and resuspension. A volume of 1–2 µL of cell suspension was placed on a flat agarose pad (1.7%) on a glass slide and imaged under phase contrast and with Texas Red and DAPI filters using a Leica DMi8 inverted microscope with a 100-fold oil-immersion lens (Leica Microsystems, Wetzlar, Germany). The images were processed using Metamorph (Universal Imaging, Bedford Hills, NY, USA).

To measure the fluorescence intensity, five images with evenly distributed cells (>100 cells/image) in each set were randomly selected for the analysis of the average fluorescence intensity using ImageJ Fiji [47]. For each image, the mean fluorescence intensity was measured in grayscale mode with the default thresholding method. The data were analyzed using Graphpad Prism version 6.0 (Graphpad Software, San Diego, CA, USA).

### 4.4. Transmission Electron Microscopy

*C. botulinum* cells were prepared and their S-layer was disrupted with 0.2 M glycine-HCl (pH 3 or 4), as described above. The samples were prepared using the method described by Hayat [48], with some modification. The samples were fixed for 24 h in 2.5% glutaraldehyde and 2% paraformaldehyde at 4 °C, and post-fixation for 1 h in 1% phosphate-buffered osmium tetroxide. Dehydration was carried out in ethanol followed by acetone. The samples were then embedded in epon and cut with a Leica ultracut UC6i ultramicrotome (Leica Microsystems). Thin sections of approximately 60–70 nm were placed on grids and visualized by a transmission electron microscope JEOL JEM-1400 (JEOL Ltd., Tokyo, Japan).

### 4.5. Flow Cytometry Separation

*C. botulinum* cells were prepared and the S-layer was disrupted as described above. Viable cell enumeration before and after S-layer disruption was performed using the three-tube MPN method [49]. All of the cell suspensions were adjusted to OD_600_ of 0.2. The cell pools were prepared by mixing equal volumes of three *C. botulinum* Group II or III cultures (CB11/1-1, 202F, and BKT2873) with *C. botulinum* Group I ATCC19397 or ATCC3502 culture, and were then incubated with mTagBFP-CBD (final concentration, 1 µM) and MTG (final concentration, 500 nM, Thermo Fisher Scientific) for 3 min at room temperature. After being washed and resuspended in an anaerobic PBS buffer, the cells were added into a MACSQuant^®^ Tyto^®^ cartridge and analyzed using the MACSQuant^®^ Tyto^®^ cell sorter (Miltenyi Biotech, Bergisch Gladbach, Germany). The fluorescence of mTagBFP-CBD and MTG was detected in the B2 channel (585/40 nm) and V1 channel (450/50 nm), respectively. Cells with a high fluorescence of mTagBFP-CBD (mTagBFP+) were gated and sorted, followed by MPN cell enumeration.

After 24 to 48 h of incubation at 30 °C or 37 °C, the cells grown in each MPN dilution series were harvested and the genomic DNA was extracted [50]. PCR detection of BoNT genes was performed as previously described using gene specific primers of *bont/A* [51], *bont/E*, *bont/F* [52], and *bont/CD* [53]. PCR products were analyzed using 2% agarose gel electrophoresis with 1 kb plus DNA ladder (New England Biolabs, Ipswich, MA, USA) as a molecular weight indicator.

### 4.6. Magnetic Separation

The coating of mCherry-CBD proteins to M-280 tosylactivated Dynabeads^®^ (Thermo Fisher Scientific) was prepared according to the manufacturer’s instructions. Briefly, 5 mg of beads were washed and re-suspended in 0.1 M borate buffer (pH 9.5) using a DynaMag™-2 magnetic tube holder. The beads were then mixed thoroughly with 100 μg of purified mCherry-CBD to give a total volume of 150 μL, and further mixed with 100 μL of 3 M ammonium sulphate buffer. The coating mixtures were incubated in a vertical rotator at 4 °C and 20 rpm for 24 h. The mCherry-CBD coated beads were washed three times with a PBST-BSA buffer (PBS containing 0.1% Tween-20 and 1% bovine serum albumin, pH 7.4) and stored anaerobically at 4 °C.

Bacterial cells were prepared and the S-layer was disrupted as described above. A total of 10^4^–10^5^ vegetative cells were mixed with ~3.3 × 10^7^ mCherry-CBD-coated M-280 beads and suspended in 100 μL of anaerobic PBST-BSA buffer. The mixtures were incubated at room temperature in a vertical rotator with rotation at 20 rpm for 30 min. After incubation, the mixtures were placed on a DynaMag™-2 magnetic tube holder for magnetic separation for 1 min. The separated M-280 beads were then washed with PBST-BSA buffer five times and re-suspended in 100 μL of anaerobic PBST-BSA buffer, followed by MPN cell enumeration. The capture efficiency for *C. botulinum* Group I and other species was determined by comparing the cell counts before and after magnetic separation. The experiments were conducted in triplicate for each species and treatment.

## Figures and Tables

**Figure 1 ijms-23-08391-f001:**
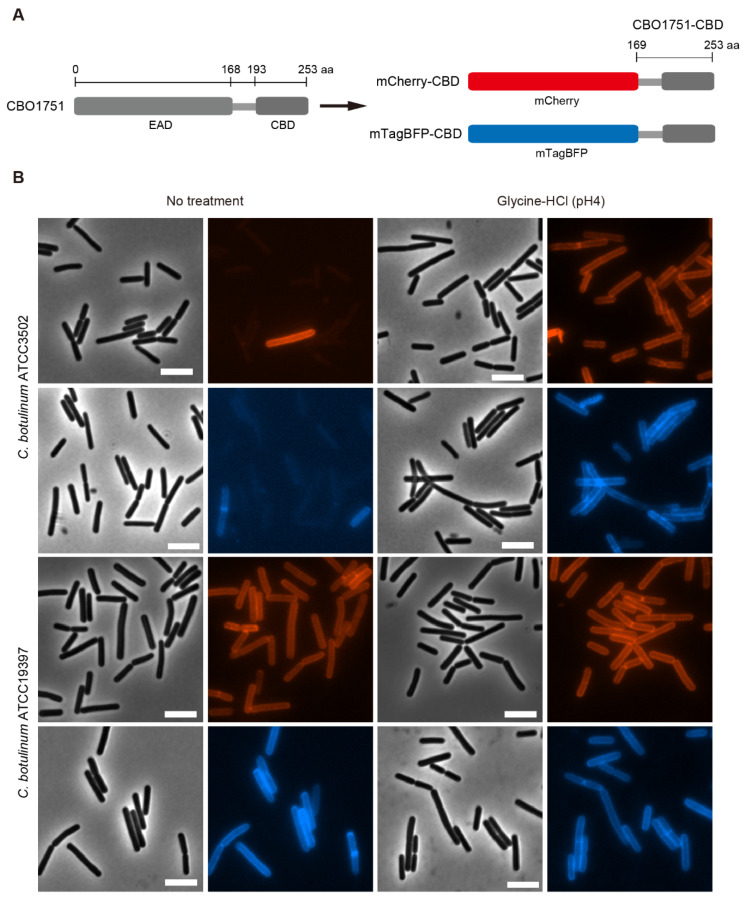
Binding of the CBO1751 cell wall binding domain (CBD) to *Clostridium botulinum* Group I cells. (**A**) Schematic representation of CBO1751 consisting of an enzymatically active domain (EAD) on the N-terminal side and a CBD on the C-terminal side, and the mCherry-CBD and mTagBFP-CBD fusion proteins. (**B**) Representative fluorescence images of *C. botulinum* ATCC3502 and ATCC19397 vegetative cells after incubation with mCherry-CBD and mTagBFP-CBD, treated or not treated with 0.2 M glycine-HCl (pH 4) for 1 min. Bars, 5 µm.

**Figure 2 ijms-23-08391-f002:**
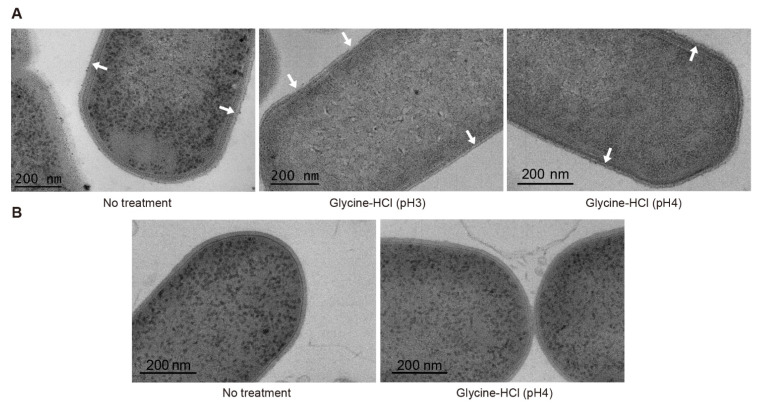
Representative TEM images of vegetative cells of *Clostridium botulinum* ATCC3502 (**A**) and ATCC19397 (**B**). The cells were treated or not treated with 0.2 M glycine-HCl (pH 3 or 4) for 1 min. White arrows indicate intact the S-layer (no treatment), broken and detached S-layer (glycine-HCl, pH 3), and granulated S-layer (glycine-HCl, pH 4).

**Figure 3 ijms-23-08391-f003:**
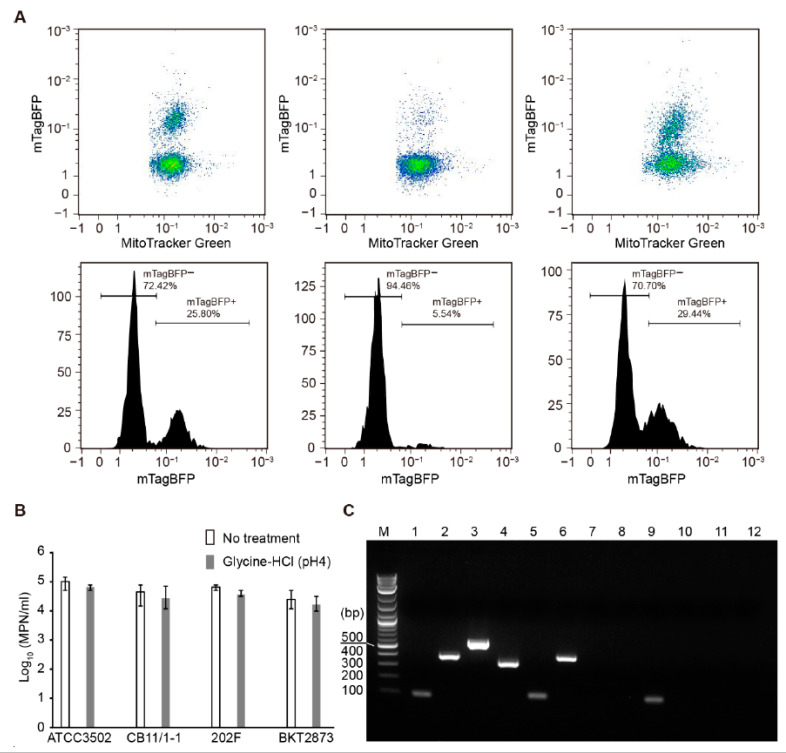
mTagBFP-CBD-based flow cytometry separation of *Clostridium botulinum* ATCC3502 and ATCC19397 from *C. botulinum* CB11/1-1, 202F, and BKT2873 strains. (**A**) Representative cytometric plots of the cell pools containing ATCC19397 with no treatment (left panel), and the cell pools containing ATCC3502 cells non-treated (middle panel) or treated (right panel) with 0.2 M glycine-HCl (pH 4) for 1 min. Gating of the subpopulations with a high intensity for mTagBFP (mTagBFP+) is indicated in the corresponding histograms. (**B**) Most probable number (MPN) enumeration of equal fractions of each *C. botulinum* ATCC3502, CB11/1-1, 202F, and BKT2873 strains that were treated or not treated with 0.2 M glycine-HCl (pH 4) for 1 min. The results are presented as the means of three replicates  ±  standard deviations. (**C**) Representative gel image of PCR products of the cell pool before sorting (lanes 1–4), and MPN dilution series (lanes 5–8, the first ten-fold dilution; lanes 9–12, the second ten-fold dilution) of the sorted cell populations of mTagBFP+. Lane M, 1 kb plus DNA ladder; lanes 1, 5, and 9, PCR products of *bont/A* (101 bp); lanes 2, 6, and 10, PCR products of *bont/E* (389 bp); lanes 3, 7, and 11, PCR products of *bont/F* (543 bp); lanes 4, 8, and 12, PCR products of *bont/CD* (327 bp).

**Figure 4 ijms-23-08391-f004:**
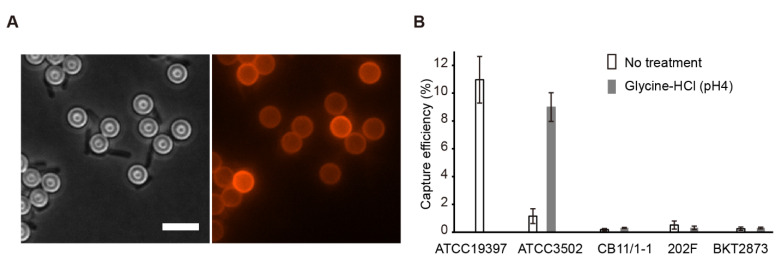
mCherry-CBD-based magnetic separation of *Clostridium botulinum* ATCC3502 and ATCC19397 from *C. botulinum* CB11/1-1, 202F, and BKT2873 strains. (**A**) Representative microscopic images of the binding of *C. botulinum* ATCC19397 cells to mCherry-CBD-coated M-280 beads. Bar, 5 µm. (**B**) Evaluation of the performance of mCherry-CBD-based magnetic separation. The capture efficiency was obtained by calculating the ratio of cell counts before and after magnetic separation. The results are presented as the means of three replicates  ±  standard deviations.

**Table 1 ijms-23-08391-t001:** Binding of mCherry-CBD to *Clostridium botulinum* and other species.

Species	Strain	Binding of mCherry-CBD ^1^	
		No Treatment	Glycine-HCl (pH 4)
*Clostridium botulinum* Group I	ATCC3502	−	+
	62A	−	+
	NCTC2916	−	+
	ATCC19397	+	+
	133-4803B	+	+
	213B	+	+
	F Langeland	+	+
*Clostridium sporogenes*	NINF45	+	+
*Clostridium botulinum* Group II	Eklund 2B	−	−
	CB11/1-1	−	−
	K126	−	−
	Eklund 202F	−	−
*Clostridium botulinum* Group III	BKT2873	−	−
*Clostridium baratii*	CCUG24033	−	−
*Clostridium butyricum*	BL86/13	−	−
*Clostridium perfringens*	ATCC13124	−	−
*Bacillus cereus*	ATCC14579	−	−
*Bacillus subtilis*	1012M15	−	−
*Listeria monocytogenes*	EGD-e	−	−
*Staphylococcus aureus*	ATCC12600	−	−
*Escherichia coli*	5 alpha	−	−

^1^ +, strong binding; −, no significant binding.

## Data Availability

All of the data analyzed or generated during the study are included in this article.

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
