# Peer review of "Specific Isolation of Clostridium botulinum Group I Cells by Phage Lysin Cell Wall Binding Domain with the Aid of S-Layer Disruption"

_ijms, 2022, doi:10.3390/ijms23158391_

Round 1

Reviewer 1 Report

Laboratory diagnostics of botulism is based on BoNTs detection in clinical samples. The confirmation of the diagnosis is achieved by identification of toxins and also isolation of viable C. botulinum cells which is very complicated and frequently lead to false negative results. The heterogenity of C. botulinum strains and their similarity to non-toxinogenic species, like C. sporogenes and C. tepidum for the Group I make an isolation process very labour and frequently very difficult to conduct. The diagnognostic of botulism frequently is carried out with using controversial MBA test, which is additionally very time consuming and extremely severe to animals. Designing of a new methods for proving the viable C. botulinum cells occurrence is desired for ensuring the safety of food and public health. One of this kind new tool has been thorougly described by authors of reviewed manuscript.  They optimized acid glycine treatment that disrupts the cell surface layer (S-layer) of C. botulinum group I, in order to obtain efficient binding of CBO1751 CBD (first antibotulinal phage lysin, previously identified by authors). This step enabled the use of CBO1751 CBD in specific identification of C. botulinum Group I clade. The authors elaborated CBO1751 CBD-based methods using flow cytometry separation and magnetic separation to 21 specifically isolate viable cells of C. botulinum Group I.

I admire the inventiveness of the authors. The isolation of C. botulinum in samples from human and animal botulism is a serious problem. I believe that this work represents an important milestone for the improvement of laboratory diagnostics in humans. I also hope, as a specialist in veterinary medicine, for a similar test for the detection of C. botulinum group III. I would like I recommend this manuscript for publication in its current form.

Reviewer 2 Report

This is an interesting and overall well-written study describing the use of the cell binding domain of a recently detected phage lysin to isolate and detect specific GI C. botulinum and C. sporogenes strains.  The idea underlying the concepts and assays described in this study are novel and innovative.  However, the broad conclusions and suggestions made by the authors are not supported by the presented evidence.  The manuscript should either be re-written to include substantial additional data, or the conclusions and statements need to be narrowed and adjusted to the specific strains investigated in the study, and limitations of the data should be discussed.  Specifically, the following needs to be addressed:

1. The Abstract is a bit difficult to read, as the sentences seem somewhat disjointed.  The goal of the study is not clear.  Similarly, the introduction doesn't completely clarify the goal and does not emphasize the underlying data from the 2020 study (Scientific Reviews, rev #10) by the same authors enough.  In line 61, authors claim great specificity of CBO1751 to GI C. botulinum but fail to mention that their own study (rev #10) also showed activity against C. sporogenes, C. baratii, and C butyricum.

2. Table 1 and supplemental Table 1:  it is unclear why the authors selected the strains shown in the table.  Why were strains from the 2020 study not included, especially C baratii and butyricum?  Why was only 1 strain of C. sporogenes included?  

3. Strains used in study:  As shown in table 1, the authors used a limited number of strains for the study, but made conclusions for the entire species.  Recent studies have also shown that several C. botulinum GI strains and several C. sporogenes strains have been misclassified.  Did the authors conduct a core genome SNP analysis of the GI and the C. sporogenes strains to ensure they are indeed falling into the respective phyla?  How representative are the selected GI strains of this species as a whole?
This is the major point were revision is needed.  Either restrict conclusions to the strains used, or provide a rationale for why the selected strains would be representative and why strains shown in the previous study to be lysed by CBO1751 were not included here.

4. The paper would be much strengthened if the cellular receptor for the CBD was determined.  This is needed to demonstrate specificity of one or several species.

5. It is not mentioned anywhere whether spores can be detected with this method.

6. Previous studies have shown that CBO1751 is heavily targeted by endogenous CRISPR systems of C. sporogenes and GI C. botulinum.  This data actually strengthens the authors and should be included in the discussion.

7. The authors should clearly distinguish their findings from the previous findings described in the 2020 paper (ref #10).  The innovative concept of using the lysin CBD to develop an isolation and detection method for food safety studies and possible research applications is quite interesting and novel, but it is not emphasized clearly in the manuscript and the reader is left wondering what hypothesis the authors intended to investigated beyond specificity of CBO1751 for GI C. botulinum, which is already published.

Round 2

Reviewer 2 Report

This revised manuscript is much improved and clarifies the scope and limitations of the study.  Since the authors have now added a paragraph about (surprising) potential weaker recognition of spores in the discussion section, the methods used for this should also be added into the methods section.  In addition, a sentence or  tow describing the results should be added to results section 2.2.  further, I would suggest to show the results as a supplementary figure, as this will need to be explored further and will be of interest to groups considering utilizing the CBD for isolation of C. bot from food matrices.  

Finally, considering that this method detects both toxic and non-toxic GI C. botulinum and C. sporogenes, an additional short discussion should be added on the utility/applicability of this method.  The authors mention potential in diagnostics.  How commonly is C. sporogenes found in diagnostic samples?  Might this method also find applicability in isolation of environmental samples and food safety studies?
